# Decentralized governance may lead to higher infection levels and sub-optimal releases of quarantines amid the COVID-19 pandemic

Adam Lampert *

Institute of Environmental Sciences, The Robert H. Smith Faculty of Agriculture, Food and Environment, The Hebrew University of Jerusalem, Rehovot, Israel

* adam.lampert@mail.huji.ac.il

## Abstract

The outbreak of the novel Coronavirus (COVID-19) has led countries worldwide to administer quarantine policies. However, each country or state independently decides what mobility restrictions to administer within its borders while aiming to maximize its own citizens' welfare. Since individuals travel between countries and states, the policy in one country affects the infection levels in other countries. Therefore, a major question is whether the policies dictated by multiple governments could be efficient. Here we focus on the decision regarding the timing of releasing quarantines, which were common during the first year of the pandemic. We consider a game-theoretical epidemiological model in which each government decides when to switch from a restrictive to a non-restrictive quarantine and vice versa. We show that, if travel between countries is frequent, then the policy dictated by multiple governments is sub-optimal. But if international travel is restricted, then the policy may become optimal.

## Introduction

The outbreak of the novel Coronavirus (COVID-19) necessitated quarantine policies, particularly during its early stages, before the vaccines were available [1–4]. While mobility restriction measures have been taken during several disease outbreaks in history [5–11], the quarantines needed for the COVID-19 pandemic were at global scales that encompass entire countries. Such quarantines have economic and social costs [12, 13], and major questions are how restrictive the quarantines should be, and what would be the right timing to release some of the mobility restrictions [1, 2, 4, 14, 15]. In practice, however, these decisions are made independently by multiple countries [16–18], or independently by multiple states in countries like the U.S., or even independently by multiple municipal authorities [19]. Each governor might incline to dictate the strategy that best serves her/his own citizens. However, in periods when the quarantines are less restrictive, travelers might transmit the disease between countries, states, and cities. Consequently, the quarantine policy in one country or state might ultimately affect the infection level in other countries and states.

In general, such decentralized governance, where decisions are made independently by each agent or governor, has a benefit: Each governor may have a better knowledge about her/

**Data Availability Statement:** The model and parameter used are fully described in the Model section. In particular, parameter values used are given with references in the "parameterization" sub-

section, and parameter values needed to regenerate Figs 2 and 3 are given in the figure legends.

**Funding:** The author received no specific funding for this work.

**Competing interests:** The authors have declared that no competing interests exist.

his own citizens' lifestyles and needs and may dictate a policy that better suits her/his own citizens. However, a decentralized policy also comes with a cost: Each government might ignore the cost borne by citizens of other countries due to international and interstate travel. In line with this argument, various previous game theory studies have suggested that agents (individuals or countries) under-invest in the prevention and control of diseases [20–26].

In this paper, we examine the case where each governor decides independently about the timing of releasing the quarantine in her/his own country or state, and we ask what the inefficiency is due to such decentralized governance. Namely, we examine how the strategies of different governments differ from the socially optimal strategy of a hypothetical centralized government that aims to maximize the welfare of all the citizens in all the countries. Specifically, we consider two countries or states, and we analyze the following three cases (Fig 1) [16–18, 27]. In case 1, we consider the scenario in which the countries have approximately the same population size and initial infection level. In case 2, we consider the scenario in which the countries have approximately the same population size, but one country experiences a more severe outbreak at a given point in time. And in case 3, we consider the scenario in which one state experiences a more severe outbreak compared to the rest of the country.

## Model

### Dynamics of the infection levels

Our model is general, but some of the assumptions and parameterizations are motivated by the COVID-19 outbreak. We consider two states or countries, 1 and 2, and in line with data about early COVID-19 outbreaks [3, 16, 18, 27–30], we assume that the number of infected people in each country is very small compared to the country's total population size. Accordingly, we consider only the dynamics of the infection level in each country, $I_i$, defined as the portion of individuals that are infected in country $i$. Namely, in contrast to traditional SIR models [31], here we consider shorter timescales during which the number of susceptible individuals is constant, which is in line with the COVID-19 data from 2020, when quarantines were common [3, 16, 18, 27–30].

We assume that, at any given time, each country is under one of two types of quarantines: restrictive or non-restrictive. The two differences between these types are that: (1) the number of infected people declines when a restrictive quarantine is administered, but increases when a non-restrictive quarantine is administered; and (2) travel between the countries occurs only if a non-restrictive quarantine is administered in both countries. Therefore, the dynamics of $I_1$ and $I_2$ depends on the type of quarantine administered in both countries over time. Specifically, when the quarantine in country $i$ is restrictive, $I_i$ decreases exponentially (increases at a negative rate $r_0 < 0$), with two exceptions: (1) if the government switches from a non-restrictive to a restrictive quarantine, $I_i$ starts to decline only after a delay of $T_{\text{delay}}$ days; (2) Even when a restrictive quarantine is administered $I_i$ does not decline below a certain threshold, $I_{\text{min}}$, characterizing the minimal level of infection in the population (see Table 1 for a list of notations). Namely, we consider a relatively restrictive quarantine, but one that still allows for some local mobility and does not eliminate the disease entirely. It follows that

$$\frac{dI_i}{dt} = \begin{cases} r_0 I_i & \text{if } I_i > I_{\text{min}} \text{ and } t_s > T_{\text{delay}} \\ 0 & \text{otherwise,} \end{cases} \tag{1}$$

where $t_s$ is the time from the day when the quarantine becomes restrictive. In turn, when a non-restrictive quarantine is administered in country $i$ and a restrictive quarantine is administered in the other country, there is still no travel between the countries, and $I_i$ increases at a

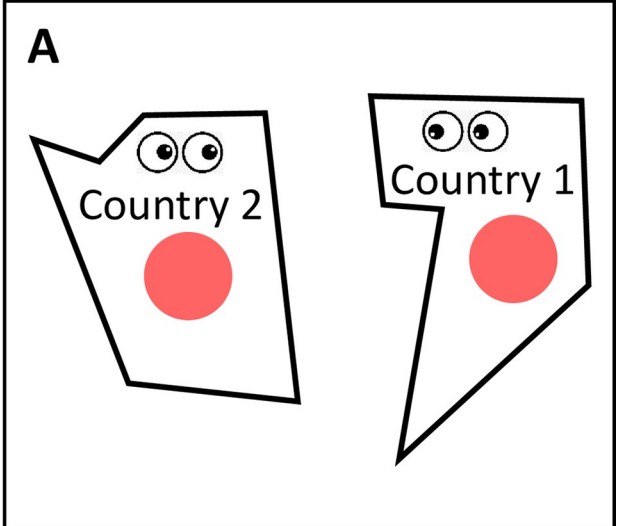

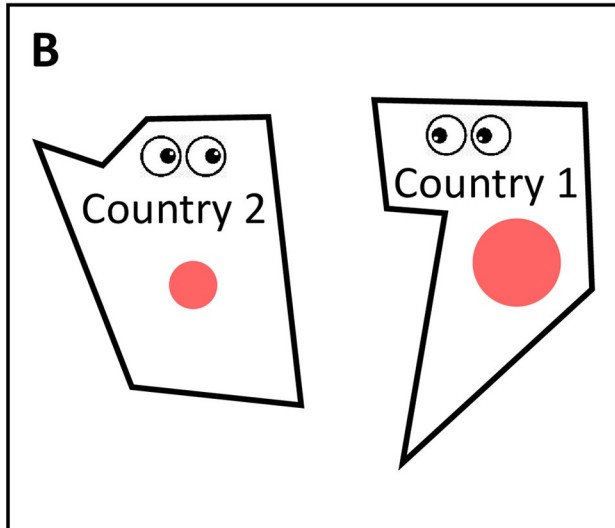

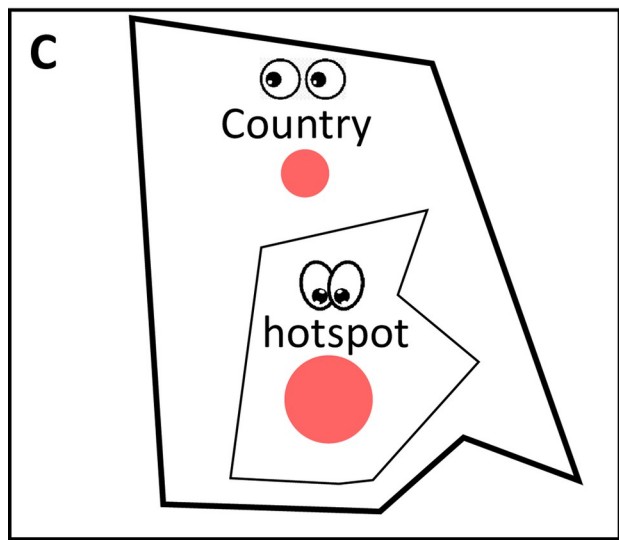

**Fig 1. Illustration of the three cases that we consider.** (A) Case 1: Both countries have the same population size and infection level. (B) Case 2: Both countries have the same population size, but country 1 has a more severe outbreak. (C) Case 3: One U.S. state is a "hotspot" and has a more severe outbreak than the rest of the U.S.

rate $r_i > 0$:

$$\frac{dI_i}{dt} = r_i I_i. \tag{2}$$

Note that $r_i$ is country-specific ($r_1$ might differ from $r_2$) and can be higher in countries in which interactions among individuals are more frequent.

Finally, when both countries are under a non-restrictive quarantine, individuals travel between the countries. We denote $\mu_{ij}$ as the portion of county $i$'s residents that travel to country $j \neq i$. We assume that a person that travels from country 1 to country 2 is still a resident of country 1 who will return to country 1. Therefore, if this traveler gets infected, this increases $I_1$ and not $I_2$, even if the transmission has occurred in country 2 [32–35] (Lagrangian approach). We also assume that the number of travelers from a country is much smaller than the number of the country's own residents ($\mu_{ij} \ll N_i$ and $\mu_{ji} \ll N_j$). This implies that

$$\frac{dI_i}{dt} = r_i I_i + \frac{r_i \mu_{ji} + r_j \mu_{ij}}{N_i} I_j. \tag{3}$$

Here, (1) the term $r_i I_i$ characterizes residents of country $i$ that become infected inside country $i$ by other residents of country $i$; (2) the term $r_i \mu_{ji} I_j / N_i$ characterizes residents of country $i$ that become infected inside country $i$ by residents of country $j$ who travel into country $i$; and (3) the term $r_i \mu_{ij} I_j / N_i$ characterizes residents of country $i$ that become infected while they travel to country $j$ (by residents of country $j$).

## Controls and objective functions

The government in each country dictates the time $T_i$ at which it switches from a non-restrictive quarantine to a restrictive one. We assume that, when $I_i$ approaches some maximum capacity, $I_i = I_{max}$, the quarantine will necessarily become restrictive in order to prevent the collapse of the health system [36]. In turn, the objective of each government is to choose $T_i$ that maximizes the portion of time during which a non-restrictive quarantine is administered in its own country, country $i$, under the assumption that the quarantine becomes restrictive

**Table 1. List of notations.**

| Symbol | Meaning |
|---|---|
| $r_0$ | Infection growth rate under restrictive quarantine |
| $r_i$ | Infection growth rate in country $i$ under non-restrictive |
| $I_{min}$ | Minimum level of infection |
| $I_{max}$ | Maximum level of infection |
| $T_i$ | The time when country $i$ chooses to switch to a non-restrictive quarantine |
| $T_{delay}$ | The time it takes for the infection level to start declining after the quarantine becomes restrictive |
| $N_i$ | Relative population size in country $i$ |
| $\mu_{ij}$ | Travel rate between countries $i$ and $j$ |
| $u_i$ | Utility of country $i$ |
| $T_i^{opt}$ | Time when country $i$ switches to a restrictive quarantine if it follows the socially optimal solution |
| $T_i^{NE}$ | Time when country $i$ switches to a restrictive quarantine if it follows the Nash equilibrium |

when $I_i$ approaches $I_{max}$. Specifically, consider a cycle during which, initially, a restrictive quarantine is administered until $t = T_i$ (stage 1); afterward, a non-restrictive quarantine is administered until $I = I_{max}$ (stage 2); and finally, the quarantine becomes restrictive again until the infection level returns to its original level $I_i(0)$ (stage 3). (In particular, if initially $I_i(0) = I_{max}$, then the duration of stage 3 is $T_{delay}$.) In turn, denote $T_i^{non}$ as the total time during the cycle when a non-restrictive quarantine is administered in country $i$ (namely, $\Delta T_i^{non}$ is the duration of stage 2). Denote $\Delta T_i^{res}$ as the total time during the cycle when a restrictive quarantine is administered in country $i$ (namely, $\Delta T_i^{non}$ is the duration of stages 1 and 3 combined). Using these notations, we define the utility of country $i$ as

$$u_i(T_1, T_2) = \frac{\Delta T_i^{non}}{\Delta T_i^{non} + \Delta T_i^{res}}. \tag{4}$$

## Optimal solution and Nash equilibrium

We find and compare two types of solutions. First, we find the socially optimal solution, $T_1^{opt}$ and $T_2^{opt}$, which maximizes the utility of the entire society in both countries combined (i.e., the combination $T_1 = T_1^{opt}$ and $T_2 = T_2^{opt}$ maximizes $N_1 u_1(T_1, T_2) + N_2 u_2(T_1, T_2)$). This solution represents the best quarantine policy in a case where a single governor makes the decision for both states. Second, we find an open-loop Nash equilibrium, which is given by the set of strategies, $(T_1^{NE}, T_2^{NE})$, for which no country can increase its own utility by unilaterally changing its strategy (i.e., $T_1 = T_1^{NE}$ maximizes $u_1(T_1, T_2^{NE})$ and $T_2 = T_2^{NE}$ maximizes $u_1(T_1^{NE}, T_2)$). Note that, in some games, a Nash equilibrium is not necessarily unique: there could be games in which multiple Nash equilibria exist, and games in which no Nash equilibrium exists [37]. Also note that, in general, the Nash equilibria in such a dynamic game depend on various assumptions about the information that each government has [38]. Here we assume that each government does not get feedback about the infection level in the other country or state and decides in advance when it releases the quarantine (open-loop Nash equilibrium [38]).

## Numerical methods

To find the socially optimal solution and the open-loop Nash equilibrium, we used a standard technique applied for games with a continuous strategy ($T_i$ varies continuously) [39]. We implemented here the algorithm for the case of two countries, and we first generate a matrix in which each cell characterizes a set of strategies, $(T_1, T_2)$. The values of $T_1$ and $T_2$ in the matrix vary between 0 and $T_{max}$, where $T_{max}$ is the time when a restrictive quarantine reduces the infection level in both countries to $I_{min}$. (It is never worthwhile to keep a restrictive quarantine for more than $T_{max}$ days.) In turn, the size of the matrix determines the resolution at which the strategies $T_1$ are $T_2$ are examined, and to generate Fig 2, we used a 200×200 matrix.

For each cell in the matrix, we calculate the utility of both countries by simulating the dynamics implied by the cell's $T_1$ and $T_2$ values. Then, to find the optimal solution, we find the cell in which the value of $N_1 u_1 + N_2 u_2$ is maximized, and we conclude that the $T_1$ and $T_2$ values of that cell are $T_1^{opt}$ and $T_2^{opt}$, respectively. In turn, we find the open-loop Nash equilibrium by calculating the best response of each country to every strategy of the other country, where the intersection of the curves characterizes the Nash equilibrium (Fig 2A, 2C and 2E).

## Parameterization

The estimations of the parameter values used for the simulation are taken from datasets and literature related to outbreaks of COVID-19 during quarantines [3, 16, 18, 27–30]. Notice that some parameter values vary significantly between countries and states, and therefore, we

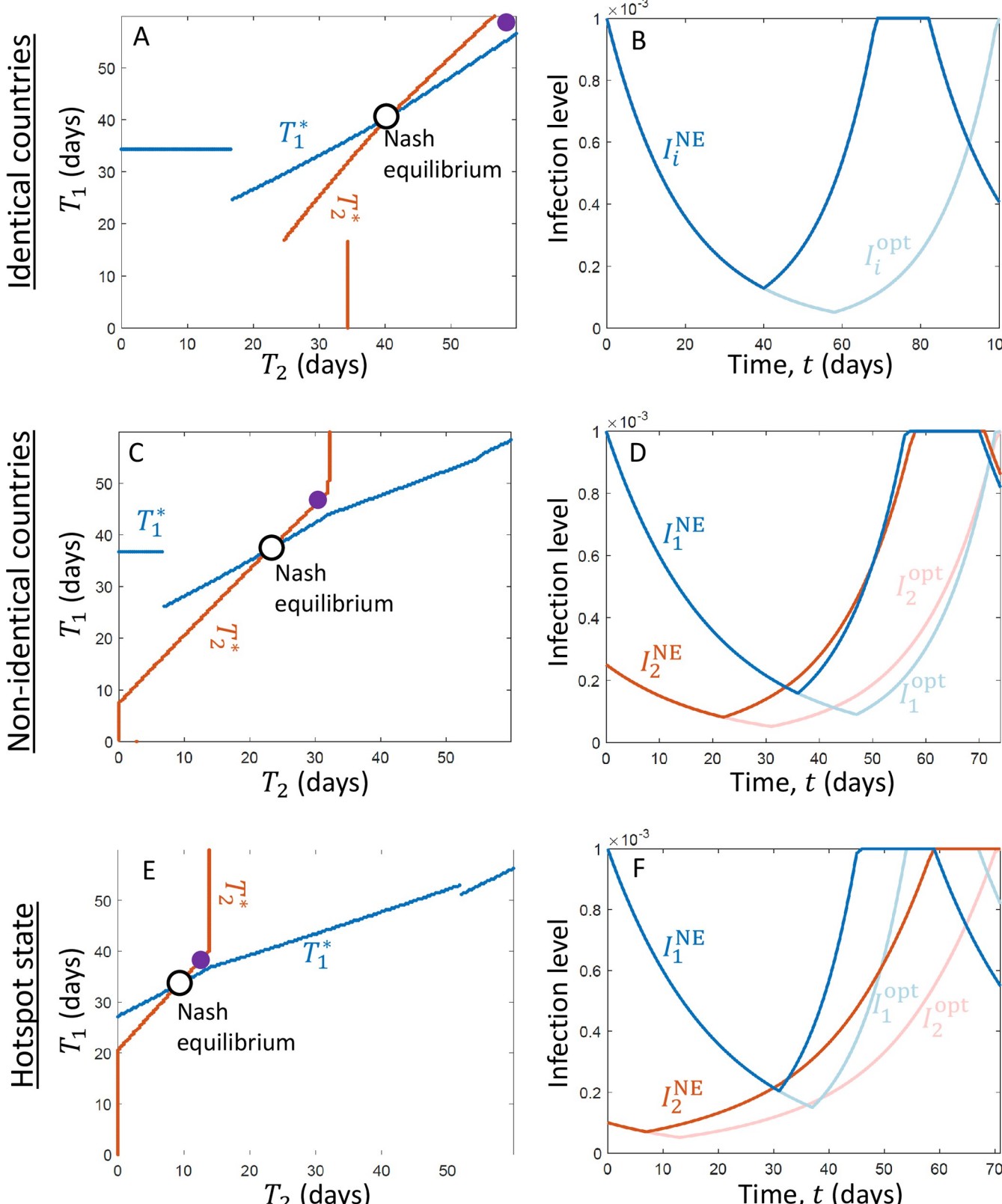

**Fig 2. The equilibrium solution dictates that governments release the quarantines earlier than optimal.** We consider the three cases illustrated in Fig 1: identical countries (A, B), non-identical countries (C, D), and a state that initially has a much higher infection level than the rest of the U.S. (E, F). The left

column (A, C, E) shows the optimal time for country 1 to release its quarantine, $T_1^*$, as a function of the time when country 2 releases its quarantine, $T_2$ (blue line). It also shows the optimal time for country 2 to release its quarantine, $T_2^*$, as a function of the time when country 1 releases its quarantine, $T_1$, on a flipped axis (mirror image, red line). The intersection of the blue and the red line indicates the open-loop Nash equilibrium. Note that a unique Nash equilibrium exists for each of the three demonstrated cases. For comparison, the purple dots show the socially optimal solution. In turn, the right column (B, D, E) shows the time evolution of the infection level in countries 1 and 2, assuming that they adopt the Nash equilibrium strategies ($I_1^{NE}$, blue line, and $I_2^{NE}$, red line), as well as the infection levels assuming that the countries adopt the socially optimal solution ($I_1^{opt}$, light blue line, and $I_2^{opt}$, light red line). In all three cases, the governments release the quarantine sooner than the optimum if they follow the Nash equilibrium. Note that in both the socially optimal solution and the Nash equilibrium, the actions of both governments tend to be synchronized: The second country that switches to a non-restrictive quarantine does so approximately when its infection level approaches that of the first country. Parameters: (all panels) $r_0 = -5\%$ (day$^{-1}$), $I_{max} = 0.1$ (% of population), $I_{min} = 0.005$ (% of population), $T_{delay}$ = 14 (days); (A, B) $r_1 = r_2 = 7\%$ (day$^{-1}$), $N_1 = N_2 = 1$, $\mu_{12} = \mu_{21} = 0.2\%$; (C, D) $r_1 = 9\%$ (day$^{-1}$), $r_2 = 7\%$ (day$^{-1}$), $N_1 = N_2 = 1$, $\mu_{12} = \mu_{21} = 0.25\%$; (E, F) $r_1 = 11\%$ (day$^{-1}$), $r_2 = 5\%$ (day$^{-1}$), $N_1 = 0.3$, $N_2 = 1$, $\mu_{12} = 0.25\% \times N_1$, $\mu_{21} = 0.25\% \times N_2$.

performed sensitivity analyses to verify that the main results are general and hold within wider parameter ranges.

Under restrictive quarantine conditions, a daily decline rate of 5% in the infection level after the first two weeks is a reasonable estimate. For example, in China, the number of infected individuals declined from ~86,000 to ~3,000 within 76 days, which implies $r_0 \approx -5\%$ per day. In turn, in several countries, it took about 14–21 days before any decline occurred following a quarantine, which suggests that considering $T_{delay} \approx 14$ days is reasonable. Next, note that $r_i$ depends on the restrictions used in a given non-restrictive quarantine, and it may vary between states and countries. We used estimates that reflect the weeks before a restrictive quarantine was administered in countries in Europe, where $5\% \leq r_i \leq 15\%$ per day is a reasonable estimate.

In turn, we assume that $I_{max}$ is given approximately by the infection level that was approached in countries in Europe before the restrictive quarantine was administered or two weeks after it was administered, where considering $I_{min} \approx 0.001\% - 0.1\%$ of the total population size is a reasonable estimate. Next, parameters like $I_{min}$ and the travel rates $\mu_{12}$ and $\mu_{21}$ are harder to estimate because they depend on the specific location and scale of the countries and states considered. Therefore, we performed sensitivity analyses and examined a variety of values (e.g., Fig 3). Finally, the ratio between $N_1$ and $N_2$ reflects the relative population sizes of the two countries: $N_1 = N_2$ characterizes the cases demonstrated in Figs 1A, 1B and 2A–2D, while $N_1 < N_2$ characterizes the case demonstrated in Figs 1C and 2E, 2F.

## Results and discussion

Our results show that, for a wide range of parameters, if each government is rational and aims to minimize the duration of the restrictive quarantine in its own country or state (decentralized governance), each government switches to a non-restrictive quarantine sooner compared to the timing that would minimize the overall duration of restrictive quarantine for both countries (Fig 2). Namely, in Nash equilibrium, the government often switches to a restrictive quarantine sooner than the socially optimal timing: $T_1^{NE} < T_1^{opt}$ and $T_2^{NE} < T_2^{opt}$. Consequently, the Nash equilibrium results in a shorter period before the infection level approaches its full capacity and the restrictive quarantine is administered again (Fig 2). Moreover, if the governance is decentralized, the solution is sub-optimal, the total amount of time during which a restrictive quarantine is administered is greater, and the average infection level is higher. This result is obtained in all the three cases we considered (Fig 1), and is consistent with the results of several previous studies, which have investigated other disease control measures taken by multiple agents, and have suggested that agents incline to under-invest in the control of the disease [20–26].

However, we also show that the difference between the Nash equilibrium and the socially optimal solution emerges only if the number of travelers between the countries, $\mu_{ij}$, is greater

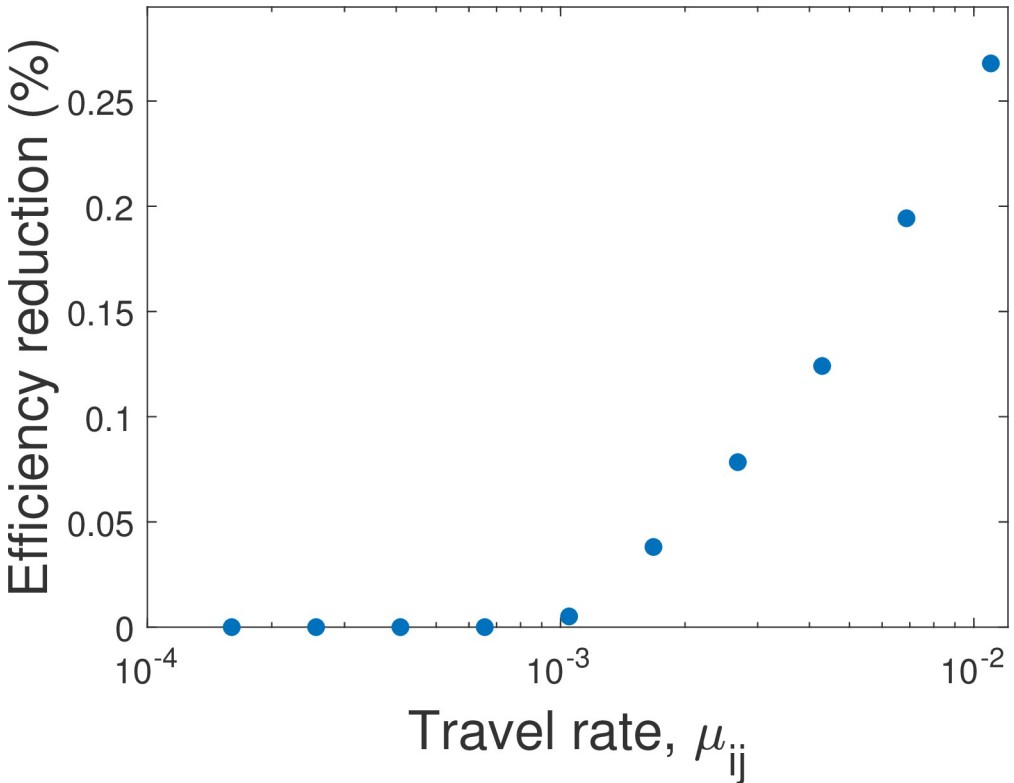

**Fig 3. Travel between states may lead to inefficient quarantine policy within each state.** Demonstrated is the increase in the relative time that a restrictive quarantine is administered (decrease in $u_i$) in a Nash equilibrium compared to the optimal solution (efficiency reduction or "cost of anarchy"; higher values are less desirable). The efficiency reduction is zero if the proportion of travelers in the population (travel rate, $\mu_{ij}$) is below a certain threshold, and increases with $\mu_{ij}$ above that threshold. Parameters other than $\mu_{ij}$ and $\mu_{ji} = \mu_{ij}$ are the same as those in Fig 2A and 2B.

than a certain threshold (Fig 3). Namely, if $\mu_{ij}$ is below the threshold, the Nash equilibrium is also the optimal solution (Fig 3). This suggests that one way to prevent the inefficiency due to decentralized governance is to restrict international or interstate travel to a low level even when the quarantines are non-restrictive. In turn, if $\mu_{ij}$ is above the threshold, the differences between the utilities obtained in Nash equilibrium and those obtained under the socially optimal solution (inefficiency; price of anarchy) increase with $\mu_{ij}$ (Fig 3).

In turn, there are two mutually dependent reasons why decentralized governance results in releasing the restrictive quarantine sooner if travel is sufficiently frequent. First, each governor ignores the damages that its own travelers inflict on other countries, and therefore, keeping a higher level of infection is perceived by the governor as less costly. Second, as a consequence of the higher infection level, (1) each country hosts more infected travelers from the other country and (2) its own travelers are hosted in countries with higher infection levels. Consequently, if the infection level in a given country is low, it increases rapidly due to travel, and therefore, it is less beneficial for a country to reduce its infection level.

Finally, note that we have made numerous simplifying assumptions in our model, which suggests that there are various future directions for examining the consequences of relaxing these assumptions. First, we consider only two countries or states, whereas in reality, these are transmissions between multiple countries and states. Second, we considered only two types of quarantine, whereas in reality, more options are available. In particular, governments can try

to administer some intermediate level of quarantine that keeps the infection at a constant level. Examining whether this policy is better than the ones that we considered is beyond the scope of this paper (see [2, 4, 36]); however, a similar kind of result might hold: Decentralized governance might maintain an infection level that is higher than the optimum. Third, we assumed that travel is allowed under non-restrictive quarantine. However, it would be interesting to examine policies that also dictate how to best integrate quarantine policy with travel policy. Specifically, further restrictions on travel might mitigate the problem (Fig 3); however, note that restrictions on travel also come with economic costs [13]. Fourth, we considered only two countries, whereas considering more countries is generally expected to increase the price of anarchy [38]. And fifth, we considered open-loop solutions in which the governments predetermine their policy, but communication among the governments might lead to the formation of agreements and coordination of a more desirable global quarantine policy.

## Acknowledgments

We sincerely thank the three reviewers for their valuable comments on the manuscript.

## Author Contributions

**Conceptualization:** Adam Lampert.

**Formal analysis:** Adam Lampert.

**Methodology:** Adam Lampert.

**Writing – original draft:** Adam Lampert.

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
