## [Decision Letter · Decision Letter 0]

14 Aug 2022

PONE-D-22-14643Decentralized governance may lead to higher infection levels and sub-optimal releases of quarantines amid the COVID-19 pandemicPLOS ONE

Dear Dr. Lampert,

Thank you for submitting your manuscript to PLOS ONE. After careful consideration, we feel that it has merit but does not fully meet PLOS ONE’s publication criteria as it currently stands. Therefore, we invite you to submit a revised version of the manuscript that addresses the points raised during the review process.

 Both reviewers agree that the paper's topic and results are interesting and novel. They also point to the need for additional clarity and rigor on the optimal control methods used. 

We look forward to receiving your revised manuscript.

Kind regards,

Ceyhun Eksin

Academic Editor

PLOS ONE

Journal Requirements:

"NO"

Reviewers' comments:

Reviewer's Responses to Questions

**Comments to the Author**

1. Is the manuscript technically sound, and do the data support the conclusions?

Reviewer #1: Yes

Reviewer #2: Partly

2. Has the statistical analysis been performed appropriately and rigorously? 

Reviewer #1: Yes

Reviewer #2: I Don't Know

3. Have the authors made all data underlying the findings in their manuscript fully available?

Reviewer #1: Yes

Reviewer #2: Yes

4. Is the manuscript presented in an intelligible fashion and written in standard English?

Reviewer #1: Yes

Reviewer #2: No

5. Review Comments to the Author

Reviewer #1: This manuscript studies a decentralized decision-making problem, in which governors of two connected countries decide when to release restricted quarantine policies for their own populations. Residents of each country also travel to the other at some fixed rates, when the quarantine policy is lifted. The authors find that, in general, decentralized policies (under a Nash equilibrium) induce more infections on average than the optimal centralized decision-making policy.

I find the topic of this study to be interesting and important. I believe the analysis and contributions can be quite valuable. The stated conclusions are clearly laid out.

My concerns are in regards to the methods, specifically the optimal control formulation. There were a few things that were not clear to me, and I thought some parts could be improved.

- What is the space of control strategies for each player? Can they implement/lift restrictive quarantines multiple times throughout the dynamics, or is it only once? It would help to formally define this.

- The application of Pontryagin’s maximum principle is not clearly detailed. If the space of control strategies is itself a timing decision, then the optimal response will be a timing decision anyway. I believe what can formally be shown here is that the optimal policy for a player is a bang-bang controller (i.e. zero-one policy), even when the space of control strategies are continuous (i.e. can select in the interval [0,1]).

- The objective function in equation (5) appears to be loosely defined. Why is it certain that the infectious level will return to its original level I_i(0)? This may follow from the setup, but please do explain this.

- Is it possible to establish some properties regarding the best-response or Nash equilibrium strategies? For example, is it always ensured that one exists? From the simulations, it appears that one does exist, and that it is unique. Is it possible for there to be multiple equilibria?

- The “brute force” numerical methods appear to work for the selected parameters, but may not generalize or scale to other scenarios. They are also not standard in optimal control or differential game studies. I would recommend using more specialized optimal control algorithms, such as forward-backward sweep methods, shooting methods, and gradient descent. See for example,

M. McAsey, L. Mou, and W. Han, “Convergence of the forward backward sweep method in optimal control,” Computational Optimization and Applications, vol. 53, no. 1, pp. 207–226, 2012.

Reviewer #2: The manuscript is based on mobility restrictions set by the governments within countries/states with the aim of maximizing the non-restrictive quarantine time for the citizens. In view of this, I would like the authors to address the following comments:

MINOR COMMENTS

There are quite a number of grammatical errors in the manuscripts which in my opinion, will give the reader a hard time to understand. For example, the following lines should be rephrased:

line 42 - a mistake I guess; "COPVID-19 outbreaks"

line 53-53 - the sentences are unclear; e.g., bracket at the beginning of a sentence

line 77 - "...between the countries..." should be "...between the countries/states..." since CASE 3 described in Figure 1C considered only a state in a country

line 87-88 - the sentences are unclear

line 161-162 - the sentences are unclear; e.g., "... are in travel..."

line 169 - a mistake I guess; "is s/he"  you meant "...if he/she...?"

line 52 & 150 - is a negative sign preceding r0 or a hyphen? If it is a negative sign, I suggest it is put in a bracket. On the other hand, if it is a hyphen, it should be replaced with a comma so as not to confuse the reader.

MAJOR COMMENTS

1. Firstly, a table describing all notations of sets, variables and parameters should be included in the manuscript; with a column showing the citations of any notation drawn from the literature.

2. "Option 1" defined in the Methods is counter-intuitive. I believe line 153 should be "...non-restrictive to restrictive...". Otherwise, Equation 1 will not hold.

3. Equation 4 should be explicitly explained. Additionally, when you consider what is written in line 171, Equation 4 appears to be incorrect as it will be impossible to arrive at Equation 3.

4. It is not clear how "Option 1", "Option 2", and "Option 3" described in the Methods section (line 144-175) are linked with the cases discussed in Figure 1 and how it was explained in the results. That is, what is the role of "Option 1/Option 2/Option 3" in Case 1/Case 2/ Case 3? Are all options considered in each of the cases or some of the options are considered? These are unclear and needs to be explicitly discussed.

5. The methods used in obtaining the "socially optimal solution" should be explained in details and discussed separately under the Method sections and also show this explicitly in Figure 2 and Figure 3 since the results described how the Nash equilibrium was compared to the socially optimal solution.

6. PLOS authors have the option to publish the peer review history of their article (what does this mean?). If published, this will include your full peer review and any attached files.

Reviewer #1: No

Reviewer #2: No

---

## [Decision Letter · Decision Letter 1]

2 Nov 2022

PONE-D-22-14643R1Decentralized governance may lead to higher infection levels and sub-optimal releases of quarantines amid the COVID-19 pandemicPLOS ONE

Dear Dr. Lampert,

Thank you for submitting your manuscript to PLOS ONE. After careful consideration, we feel that it has merit but does not fully meet PLOS ONE’s publication criteria as it currently stands. Therefore, we invite you to submit a revised version of the manuscript that addresses the points raised during the review process.

Thank you for addressing the comments of the reviewers. Unfortunately, I had to find a new reviewer for this round because one of the reviewers from the previous round was unavailable. The new reviewer has additional set of concerns regarding generalizability of the model and validity of the proposed dynamics in epidemics. These comments are valid and merit another revision. Addressing this reviewer's comments satisfactorily would improve the manuscript significantly and elevate its potential impact.

We look forward to receiving your revised manuscript.

Kind regards,

Ceyhun Eksin

Academic Editor

PLOS ONE

Journal Requirements:

Reviewers' comments:

Reviewer's Responses to Questions

**Comments to the Author**

1. If the authors have adequately addressed your comments raised in a previous round of review and you feel that this manuscript is now acceptable for publication, you may indicate that here to bypass the “Comments to the Author” section, enter your conflict of interest statement in the “Confidential to Editor” section, and submit your "Accept" recommendation.

Reviewer #1: All comments have been addressed

Reviewer #3: (No Response)

2. Is the manuscript technically sound, and do the data support the conclusions?

Reviewer #1: Yes

Reviewer #3: Partly

3. Has the statistical analysis been performed appropriately and rigorously? 

Reviewer #1: Yes

Reviewer #3: I Don't Know

4. Have the authors made all data underlying the findings in their manuscript fully available?

Reviewer #1: Yes

Reviewer #3: Yes

5. Is the manuscript presented in an intelligible fashion and written in standard English?

Reviewer #1: Yes

Reviewer #3: Yes

6. Review Comments to the Author

Reviewer #1: (No Response)

Reviewer #3: The paper analyzes the role of decentralized control of epidemic spreading over sub-populations through lock-down interventions. Through constructing the cost function as a function of the intervention, the paper studies the optimal- and sub-optimal intervention strategies. Analysis of a Nash equilibrium of the model is given. I have the following major concerns about the paper.

1) My biggest concern is that the paper studies an epidemic spreading over a two-country spreading model. The results are developed on the two-country spreading model. In real-world scenarios, we might need a generalized multi-population spreading model. How can the results developed in this work be applied to a more generalized multi-population spreading model, in terms of theoretical analysis of the equilibrium, comparison of the optimal intervention period, and the numerical analysis of the solution?

2) Starting from line 29, the paper claims that “each government might ignore the cost

borne by citizens of other countries due to international and interstate travel”. In reality, e.g., during the COVID-19 pandemic, it is more common for a country to only considers the impact/cost of the intervention within the country itself, instead of the what the intervention will bring to other countries. The paper should give some solid examples to show the reason why a country should consider the impact of the intervention on other countries.

3) The paper gives three assumptions from line 38 to line 41. All assumptions assume both countries have around the same population, which are strong assumptions in real world. Is there a way to relax the population assumptions? In addition, under the assumption that both countries have around the same population, is it necessary that the cost function given in line 96, i.e., 11(1, 2) + 22(1, 2)) contains N1 and N2?

4) In line 60, the paper assumes that the infected population of each country cannot be lower than a threshold. The assumption is in general good. However, extremely strict lock-down can eradicate the epidemic completely, e.g., China did it during 2021. It might be better to further clarity the definition of restrictive quarantine and non-restrictive quarantine in this paper. If the restrictive quarantine is as restrictive as China did before, it is possible that the infected population can reach zero for a while.

5) Is equation (3) a popular way of modeling epidemic transmission? If so, please add a couple of reference. From my own understanding, the transmission and infection are usually model through the Law of mass action, given by nonlinear dynamics. However, equation (3) uses a group of linear dynamics to capture the change of the infected population, which may not be well-accepted in the epidemiology and disease spreading area.

6) The descriptions of strictive and non-strictive quarantine from 89 to 92 do not match the introductions of these three stages from line 82 to line 88. It is confusing that the definition of Stage 1 means the beginning of a stage or the end of a stage.

7) In the section “Optimal solution and Nash equilibrium”, the paper gives the socially optimal solution. Can the paper give more intuitive explanations about the solutions in real world implementation? The paper also mentioned that there could be cases in which multiple Nash equilibria exist, and cases in which no Nash equilibrium exists. Can the paper give more detailed information on when and how these cases appear under the current setting?

8) In the numerical simulation, how scalable the proposed method can be? The paper considers a two-country case. What if we need to consider twenty or even more countries? Can the numerical method be implemented to large-scale networked spreading models?

7. PLOS authors have the option to publish the peer review history of their article (what does this mean?). If published, this will include your full peer review and any attached files.

Reviewer #1: No

Reviewer #3: No

---

## [Author Response · Author response to Decision Letter 1]

11 Nov 2022

Please find attached Response to Reviewers letter at the end of the manuscript.

---

## [Decision Letter · Decision Letter 2]

1 Dec 2022

Decentralized governance may lead to higher infection levels and sub-optimal releases of quarantines amid the COVID-19 pandemic

PONE-D-22-14643R2

Dear Dr. Lampert,

We’re pleased to inform you that your manuscript has been judged scientifically suitable for publication and will be formally accepted for publication once it meets all outstanding technical requirements.

Kind regards,

Ceyhun Eksin

Academic Editor

PLOS ONE

Additional Editor Comments (optional):

This revision addressed all of the reviewers' comments. Thank you,

Reviewers' comments:

Reviewer's Responses to Questions

**Comments to the Author**

1. If the authors have adequately addressed your comments raised in a previous round of review and you feel that this manuscript is now acceptable for publication, you may indicate that here to bypass the “Comments to the Author” section, enter your conflict of interest statement in the “Confidential to Editor” section, and submit your "Accept" recommendation.

Reviewer #1: All comments have been addressed

Reviewer #3: All comments have been addressed

2. Is the manuscript technically sound, and do the data support the conclusions?

Reviewer #1: Yes

Reviewer #3: Yes

3. Has the statistical analysis been performed appropriately and rigorously? 

Reviewer #1: Yes

Reviewer #3: Yes

4. Have the authors made all data underlying the findings in their manuscript fully available?

Reviewer #1: Yes

Reviewer #3: Yes

5. Is the manuscript presented in an intelligible fashion and written in standard English?

Reviewer #1: Yes

Reviewer #3: Yes

6. Review Comments to the Author

Reviewer #1: The authors have addressed previous concerns regarding the setup of the problem and methods used for analysis.

I recommend publication of this version of the paper. It provides interesting insights into decentralized decision-making for quarantine policies. While the methods appeared ad-hoc (numerical brute-force search), they sufficed for solving the problem setup with two country setting. Methods that can be generalized may be necessary when studying more than two countries.

Reviewer #3: The paper has addressed all my comments. I recommend the publication of the paper in its current version.

7. PLOS authors have the option to publish the peer review history of their article (what does this mean?). If published, this will include your full peer review and any attached files.

Reviewer #1: No

Reviewer #3: No

---

## [Editor Report · Acceptance letter]

5 Dec 2022

PONE-D-22-14643R2 

Decentralized governance may lead to higher infection levels and sub-optimal releases of quarantines amid the COVID-19 pandemic 

Dear Dr. Lampert:

I'm pleased to inform you that your manuscript has been deemed suitable for publication in PLOS ONE. Congratulations! Your manuscript is now with our production department. 

Kind regards, 

on behalf of

Dr. Ceyhun Eksin 

Academic Editor

PLOS ONE